# DISCO: DYNAMIC SCHEDULING FOR CPU OFFLOAD IN ML WORKLOADS

## ABSTRACT

An obvious way to alleviate memory difficulties in GPU-based ML workloads is via *CPU offload*, where data are moved between GPU and CPU RAM. While CPU offload is useful, it can greatly slow down a computation due to the relatively slow transfer rate between CPU RAM and GPU RAM. To address this, overlapping memory transfer and compute is a necessity. In this paper, we present a unique approach to CPU offload in ML workloads, called DISCO (**D**ynam**I**c **S**cheduling for **C**pu **O**ffload). DISCO views an ML workload as a fine-grained dataflow graph. Operations in the graph are individual kernel calls to be run on a specific GPU, CPU-to-GPU transfers, GPU-to-CPU transfers, and GPU-to-GPU transfers. DISCO makes use of a work-conserving, dynamic scheduler to asynchronously execute the operations in the graph, whenever the underlying resource is available and the system can be sure that executing the operation cannot violate the correctness of the computation. In this way, DISCO ensures that all resources–GPUs, CPU–to–GPU bus–are fully utilized.

## 1 INTRODUCTION

One of the key ideas for preventing out-of-memory (OOM) errors in ML workloads is utilizing inexpensive CPU RAM to augment expensive GPU RAM. Most proposals for CPU offload leverage the fact that modern ML models, such as large transformers, process data in a levelwise fashion (Ren et al., 2021; Aminabadi et al., 2022; Sheng et al., 2023). As a layer is processed on a GPU server, the model weights associated with a given layer can be loaded from CPU RAM to GPU RAM in a "just in time" fashion. Once the weights associated with a level are loaded into GPU RAM, the computations associated with the level can be executed.

The problem with this approach when realized on a multi-GPU server is that these various bulk operations—such as moving the weights associated with a layer into the RAM of the various GPUs on the server, or processing a transformer layer using those GPUs once the weights have been loaded—are fully synchronous when implemented on top of modern ML frameworks such as PyTorch or JAX. In these existing "bulk synchronous" systems, all operations in a layer operate synchronously. A layer cannot begin processing until all the weights have been transferred. This means that GPUs may sit idle, even though *some* of the weights associated with a level are ready to be used.

In this paper, we present a very different approach to CPU offload in ML workloads, called DISCO (**D**ynam**I**c **S**cheduling for **C**pu **O**ffload). DISCO views an ML workload as a fine-grained dataflow graph. Operations in the graph are individual kernel calls to be run on a specific GPU, CPU-to-GPU transfers, GPU-to-CPU transfers, and GPU-to-GPU transfers. DISCO makes use of a work-conserving dynamic scheduler (Kleinrock, 1965) to asynchronously execute the operations in the graph. Because the DISCO scheduler is "work conserving"—it will never let a resource such as a CPU-to-GPU PCIe bus go idle if it can be used without violating correctness constraints—DISCO can reduce the runtime of ML workloads that must perform CPU offload.

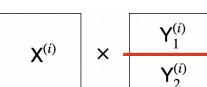

Figure 1: A tensor-parallel decomposition of a layer in a neural network.

For an example of how DISCO can be faster than classical, levelwise processing, consider the problem of running a simple ML workload that, at level $i$ of a neural network, performs the matrix multiplication

$$\mathbf{X}^{(i+1)} \leftarrow \mathbf{X}^{(i)} \times \mathbf{Y}^{(i)}.$$

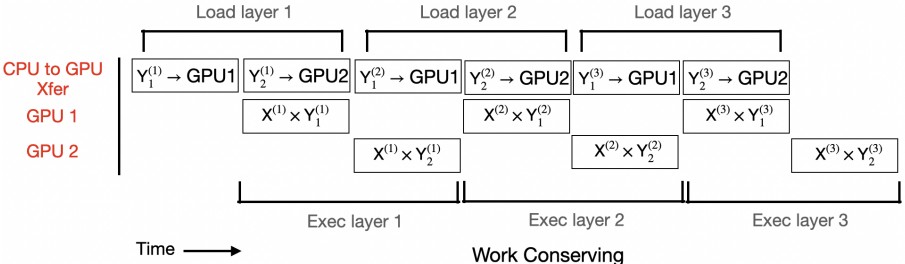

Figure 2: Levelwise CPU-to-GPU transfer. Loading and execution of layers does not overlap.

Figure 3: A work-conserving schedule. Operations such as CPU-to-GPU transfer and GPU kernels are executed as soon as all dependencies have been satisfied. As a result, the bus connecting the CPU and GPU is fully utilized.

$\mathbf{X}^{(i)}$ is a matrix holding the activations for a list of data points at level $i$ in a neural network, and $\mathbf{Y}^{(i)}$ is a weight matrix at level $i$ in the network. We wish to run this on a two-GPU system, using a tensor-parallel decomposition (see Figure 1). Assume that we do not have enough GPU RAM to store each $\mathbf{Y}^{(i)}$ on the GPU, and so these matrices must be offloaded to the CPU. As memory is limited, we can only store one $\mathbf{Y}_j^{(i)}$ on the GPU at a time. Further, assume for simplicity that a matrix multiplication and the CPU-to-GPU transfer take roughly the same amount of time.[1]

As illustrated in Figure 2 and Figure 3, a neural network with $n$ layers requires $3n$ time units under levelwise processing and $2n + 1$ time units under a work-conserving schedule. When $n$ is large, levelwise processing is 50% slower than a work-conserving, dynamic scheduler in this case. While bulk-synchronous, levelwise processing that must run the entire layer at once cannot overlap the execution of a layer with the transfer of a layer from CPU-to-GPU, a work-conserving scheduler can run CPU kernels and CPU-to-GPU transfers as soon as all dependencies are satisfied. For example, it can execute $\mathbf{X}^{(1)} \times \mathbf{Y}_1^{(1)}$ as soon as $\mathbf{Y}_1^{(1)}$ is on GPU1. And it can commence the transfer of $\mathbf{Y}_1^{(2)}$ as soon as the aforementioned multiplication is done and the space utilized by $\mathbf{Y}_1^{(1)}$ is no longer needed.

While it may be possible to engineer a solution that works well in this case, not all ML workloads are as simple as this example. It may be possible to store most of a layer in GPU RAM, and so it is not necessary to load *all* weights associated with a layer, but just *some* of them. There may be arbitrarily complicated dependencies in the computation, so that certain intermediate results will not be used for some time, and thus they should be offloaded from GPU to CPU RAM to save space, whereas others will be used immediately, and should not be offloaded. Some computations may not have a nice levelwise structure.

The key innovation of DISCO is its combination of a pre-computed memory access plan called a MEMGRAPH that allows a system based on a work-conserving, dynamic scheduler, to handle all of these cases. A MEMGRAPH is a dependency graph where vertices represent tasks (such as the execution of a GPU kernel to perform a small part of attention computation in a layer of a large language model) and edges represent data or memory dependencies. Any execution order that respects the dependencies in the MEMGRAPH is valid, and tasks are dispatched at any time that their dependencies have been met and the appropriate resources are free. Thus, depending upon

---

[1] Given a modern GPU and PCIe bus, the time required to transfer a large, roughly square matrix to GPU RAM, and the time required to multiply that matrix by another, similarly-sized matrix on a GPU, will be of the same order of magnitude. The larger the matrix, the more costly, relatively speaking, the multiplication becomes.

when events (kernel calls, transfers) finish, two executions of the same MEMGRAPH may lead to different sequences of operations being executed on a GPU, or different sequences of tensors being paged to CPU RAM. However, the dependencies in the MEMGRAPH are such that the final output is always *correct*, no matter the execution order. Because operations can be dispatched whenever the dependencies are fulfilled and are not constrained to any specific ordering, it lowers the chance that any GPU will be stalled waiting for a memory transfer to complete. If one task cannot run due to an un-met dependency in the MEMGRAPH, it is possible that there is another task that *can* run.

The technical challenge is how to effectively build a MEMGRAPH with as few dependencies as possible, to allow the runtime as much freedom as possible to dispatch operations so that it is never blocked, waiting for a memory transfer to complete. DISCO builds a MEMGRAPH by simulating an execution of the computation, mapping tensors to GPU memory locations and, when necessary, adding edges that represent memory dependencies, as well as `offload` and `reload` operations.

## 2 RELATED WORK

There are two approaches taken by systems dealing with limited GPU memory. Some, like DISCO, accept an abstracted version of a generic GPU computation. Other systems are more specifically targeted to certain categories of models, optimization algorithms, or to specific tasks such as training or inference.

Using the first, more general approach, are systems that accept a generic dataflow graph and, like DISCO, plan for execution in limited memory: `pofo` (Beaumont et al., 2021), AutoTM (Hildebrand et al., 2020), SwapAdvisor (Huang et al., 2020), Checkmate (Jain et al., 2020), Capuchin (Peng et al., 2020), and POET (Patil et al., 2022) all assume an input dataflow graph for a machine learning computation, and then plan for execution in limited memory. Checkmate considers only tensor re-materialization, whereas POET, `pofo`, and Capuchin consider re-materialization and offload; AutoTM and SwapAdvisor consider only offload.

The more targeted approach is taken by the DeepSpeed project (Deepspeed) and the various ZeRO optimizations. For transformers and other, similar models, DeepSpeed inference (which includes ZeRO-Inference) (Aminabadi et al., 2022) has two key ideas. First, DeepSpeed inference "offload[s] some activation from GPU to CPU memory while not in use." Second, DeepSpeed inference "pins the model weights either in DRAM (if large enough) or NVMe, and streams each layer into GPU memory for computation when needed." FlexGen (Sheng et al., 2023) seeks to use a variety of methods to speed transformer inference given limited hardware, including model weight offload to CPU, quantization (Yao et al., 2022; Frantar et al., 2022), and sparse attention (Child et al., 2019). The latter two ideas are orthogonal to the ideas in this paper. For CPU offload, FlexGen optimizes a "zig-zag" block scheduling that works through transformer layers and sequences in the batch, offloading and reloading the KV-cache (Pope et al., 2023) and model weights. PagedAttention (Kwon et al., 2023) deals with low memory utilization in transformers, developing a paging system for the KV-cache.

ZeRO-Offload (Ren et al., 2021) is a comprehensive solution for limited-memory training that can be seen as primarily using CPU RAM for running the ADAM optimizer, moving weights to GPU RAM on a carefully-controlled schedule. ZeRO-Offload is an enhancement on ZeRO (Rajbhandari et al., 2020), which is designed to be memory-efficient, partitioning both the optimizer and the data across multiple GPUs. ZeRO-Infinity (Rajbhandari et al., 2021) is similar, and includes a CPU offload engine, as well as tiling of operators to utilize the RAM of multiple GPUs.

## 3 TASKGRAPHS AND MEMGRAPHS IN DISCO

DISCO takes as input a TASKGRAPH. A TASKGRAPH is a dataflow graph (a directed, acyclic graph) that describes how to perform multi-GPU computations. In a TASKGRAPH, edges represent data flow, and vertices represent operations over tensors. A vertex without any inputs (called an *input vertex*) is associated with an input tensor. An operation associated with a non-input vertex may be either a kernel call that is to be executed on a specific GPU, or a GPU-to-GPU data transfer.

DISCO is agnostic as to how the TASKGRAPH is created; it could, for example be created using a framework such as FlexFlow (Jia et al., 2019) or Alpa (Zheng et al., 2022). Consider a matrix

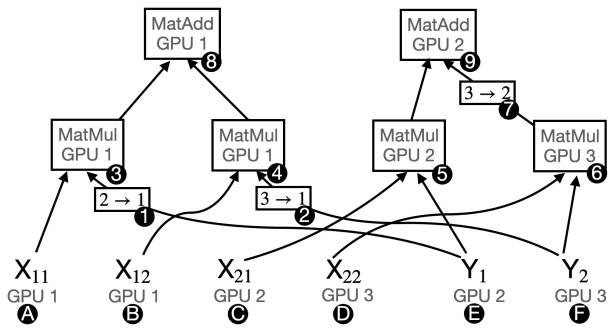

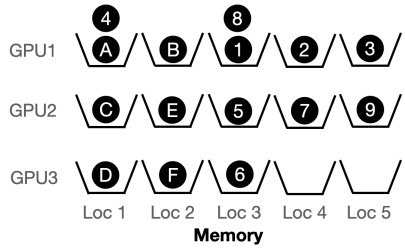

Figure 5: Possible mapping of the output of all of the operations in Figure 4 to memory locations.

Figure 4: Example TASKGRAPH consisting of six GPU kernel calls and three GPU-to-GPU transfers.

multiplication $\mathbf{X} \times \mathbf{Y}$ and assume we wish to execute this matrix multiplication on three GPUs. To produce a TASKGRAPH, a framework such as FlexFlow may choose to decompose this matrix multiplication as depicted in Figure 6, perhaps corresponding to the TASKGRAPH of Figure 4.

Given such a TASKGRAPH, DISCO first compiles the TASKGRAPH into a MEMGRAPH, which it will eventually execute. Like a TASKGRAPH, a MEMGRAPH is also a directed acyclic graph. Every vertex in the original TASKGRAPH will be present in a corresponding MEMGRAPH. Further, the compilation process may add additional `offload` and `reload` operations that move memory from GPU RAM to CPU RAM, and vice versa. During the compilation process, the output associated with every vertex in the MEMGRAPH is mapped to a memory location. Unlike the

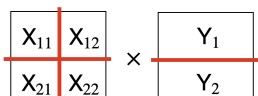

Figure 6: A decomposition of matrix multiplication.

input TASKGRAPH, the MEMGRAPH is not a dataflow graph; it is a dependency graph. If there is an edge from $v_1$ to $v_2$, it means that $v_2$ depends on $v_1$ and $v_2$ may not execute until after $v_1$ has been executed. In a MEMGRAPH, there are two types of dependencies. One is a data dependency, which is inherited from the TASKGRAPH (or is created via the addition of an `offload` or `reload`; see below). The second is a memory dependency, which is added to ensure that there are no race conditions in the graph. A *race condition* occurs when there is some vertex for which two valid executions of the graph may produce a different output. This can happen when two vertices write to the same memory location, and it is possible for a third vertex to read either output, depending upon the execution order.

Let us illustrate a possible compilation of the TASKGRAPH of Figure 4 to a MEMGRAPH. Imagine that our three GPUs each have five memory locations, and for simplicity, each tensor is the same size and occupies exactly one memory location. During compilation, the tensor associated with the output of each operation in the TASKGRAPH is assigned to a memory location, as depicted in Figure 5. GPU 1 must deal with seven tensors total (two input tensors and five additional tensors that are created via the execution of some operation), and we cannot fit all seven of those tensors in memory, given our five locations. Thus, the tensors output by operations A and 4 are both mapped to GPU1-Loc1, and the tensors output by operations 1 and 8 are both mapped to GPU1-Loc3.

A corresponding MEMGRAPH is shown in Figure 7. Note that two new edges representing memory dependencies have been added. These edges guarantee that the graph is free of race conditions. Specifically, a graph will be free of race conditions, if, whenever the outputs of vertices $v_1$ and $v_2$ have both been mapped to the same memory location, either $v_1$ *safely overwrites the result of* $v_2$, or $v_2$ *safely overwrites the result of* $v_1$. We say that "$v_1$ safely over-writes the result of $v_2$" if and only if, for every $v_3$ that consumes the output of $v_2$, there is a memory dependency from $v_3$ (or some descendent of $v_3$) to $v_1$ (or to some ancestor of $v_1$). Why? If $v_1$ is to safely over-write the result of $v_2$, we need to ensure that $v_1$ cannot execute until all of the consumers of $v_2$ have finished execution—such memory dependencies ensure this.

For example, from Figure 5 we see that the output of vertex 4 is mapped to the same location as the output of vertex A. In the associated MEMGRAPH of Figure 4, to ensure that 4 safely over-writes the result of A, we add a memory dependency from 3 (the only consumer of A) to 4. From Figure 5 we also see that the results of 1 and 8 are mapped to the same location. To ensure that 8 safely over-writes the result of 1, we add a memory dependency from 3 (the only consumer of 1) to 8. Note

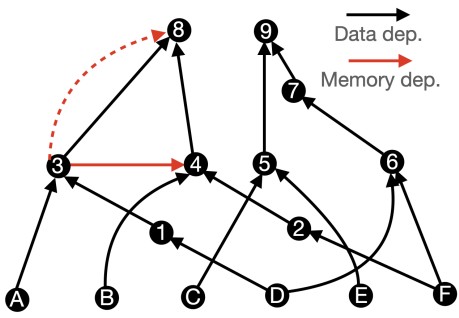

Figure 7: MEMGRAPH corresponding to Figure 4.

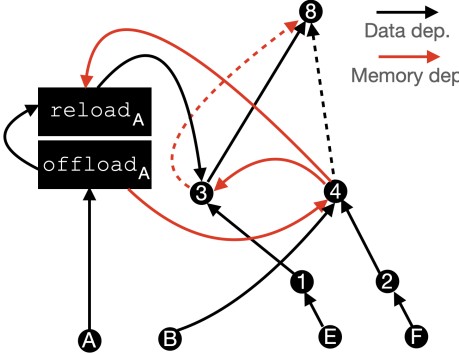

Figure 8: MEMGRAPH with less GPU RAM.

that this memory dependency is shown as a dashed line; this indicates that it is *superfluous*, as there is already a data dependency from 3 to 8, so this memory dependency is not needed for correctness.

Things can become more intricate if the memory is more constrained. Consider the case where we have only four memory locations on each GPU, and we wish to compile the same TASKGRAPH. One possible mapping of the vertices TASKGRAPH of Figure 4 to memory locations for GPU 1 is shown in Figure 9; the associated TASKGRAPH is shown in Figure 8. Note in particular the addition of an `offload`-`reload` pair. Both the `offload` and the `reload` are new operations that are added to the MEMGRAPH during compilation, to facilitate execution in memory-constrained scenarios. We can always compile $v_1 \rightarrow v_2$ in a TASKGRAPH to $v_1 \rightarrow \texttt{offload}_{v_1} \rightarrow \texttt{reload}_{v_1} \rightarrow v_2$ in a MEMGRAPH. After the $\texttt{offload}_{v_1}$, the result of $v_1$ takes up no GPU memory, but it cannot be used until the $\texttt{reload}_{v_1}$, where it is again mapped to a GPU memory location. The reason for the inclusion of the `offload`-`reload` pair in this case is that it allows the result of A to be removed from GPU RAM for a time. Thus, vertex 4 can execute and write its result on top of the result of A, which is later reloaded so that vertex 3 can be executed.

In Figure 9 we see that there are four pairs of vertices whose results are mapped to the same GPU memory locations, and so memory dependencies must have been added to the MEMGRAPH to ensure that there are no race conditions. Consider A and 4, which are both mapped to GPU1-Loc1. To ensure that 4 safely over-writes the result of A, we have a memory dependency from the $\texttt{offload}_{v_1}$ (the only consumer of A) to 4. Or, consider $\texttt{reload}_{v_1}$ and 2, which are both mapped to GPU1-Loc4. To ensure that the $\texttt{reload}_{v_1}$ safely over-

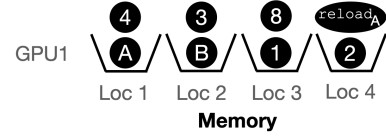

Figure 9: Possible mapping of tensors to GPU RAM.

writes the result of 2, there is a memory dependency from the only consumer of 2 (vertex 4) to the $\texttt{reload}_{v_1}$.

## 4    BUILDING A MEMGRAPH

The key technical question we address in this paper is: How to construct a MEMGRAPH from a TASKGRAPH? The primary requirement for the compilation process is *correctness*. Correctness requires that (a) every data dependency present in the TASKGRAPH is also present in the MEMGRAPH, *or* is replaced with a sequence of `offload`-`reload` operations;[2] (b) there are no race conditions in the MEMGRAPH; (c) the MEMGRAPH has no cycles. In addition, it is desirable for the MEMGRAPH to be performant. A MEMGRAPH will not be performant if memory dependencies severely constrain the execution order of vertices. Such constraints may reduce parallelism and GPU utilization.

Our basic tactic during compilation is to rely on a simulated execution of the TASKGRAPH to generate the MEMGRAPH. Given a serial ordering of the vertices in the TASKGRAPH that respects all dependencies (so that if $v_1 \rightarrow v_2$ is in the TASKGRAPH, $v_1$ is before $v_2$ in the ordering) we

---

[2]So, for example, if $v_1 \rightarrow v_2$ is present in the TASKGRAPH, we may have $v_1 \rightarrow \texttt{offload}_{v_1} \rightarrow \texttt{reload}_{v_1} \rightarrow v_2$ in the MEMGRAPH

BUILDMEMGRAPH: **Inputs**: TASKGRAPH, sorted list of TASKGRAPH vertices $V = \langle v_1, v_2, ..., v_n \rangle$; **Outputs**: MEMGRAPH, GPU memory location $v_i.\texttt{loc}$ for $i \in \{1...n\}$

```
Evicted←{}; execHzn←1; allocHzn←1;
while execHzn ≤ n do
  if allocHzn ≤ n and (v_allocHzn.loc ← simMalloc(v_allocHzn)) ≠ −1 then
    /*successfully allocated space for future result*/
    allocHzn += 1
  else if allocHzn = execHzn then
    /* unable to allocate for next execution w/o evict*/
    v_allocHzn.loc←simMallocOffld(v_allocHzn)
    allocHzn += 1
  else
    /* simulate execution of the next vertex */
    /* first, compute set of vertices exec depends on */
    Deps←{v s.t. edge v→v_execHzn ∈ TASKGRAPH}
    for v ∈ Deps do
      /* reload dependency if evicted */
      if v ∈ Evicted then
        v.loc←simMallocForceReld(v)
      end if
      /* if dependency won't be used again, free it */
      if not ∃(fut > execHzn s.t. edge v→v_fut ∈ TASKGRAPH) then
        simFree(v)
      end if
      add edge v→v_execHzn to MEMGRAPH
    end for
    execHzn += 1
  end if
end while
```

Figure 10: Building a MEMGRAPH via execution simulation.

simulate its execution, making calls to special variants of `malloc` and `free` that do not actually allocate GPU RAM, but instead maintain a map of used and free RAM slots on the GPU that is the target of the compilation. These implementations also maintain a history of which tensors occupied which positions in simulated GPU RAM, to correctly generate memory dependencies. As the simulation runs, the MEMGRAPH is constructed. Calls to the special `malloc` implementations associate MEMGRAPH vertex outputs to GPU memory locations (effectively producing the mappings depicted in Figure 5 and Figure 9). Whenever a call to the `malloc` variant fails because there is not enough GPU RAM, an `offload` vertex must be added to the MEMGRAPH. Whenever it is time to simulate the execution of a TASKGRAPH vertex but one of the inputs is not in the simulated GPU RAM, then a memory location for the corresponding `reload` vertex is allocated, and a data dependency on that `reload` is added to the MEMGRAPH.

As the simulation runs, there are two *horizons*, or counters that mark progress through the serialized TASKGRAPH. The first is the `allocHzn`. Every vertex in the TASKGRAPH that is older than the `allocHzn` has had a space allocated for it. The second is the `execHzn`. Every vertex in the TASKGRAPH that is older than the `execHzn` has been "run" according to the simulation. To ensure a high-quality MEMGRAPH, our compilation algorithm greedily tries to push the `allocHzn` as far as possible past the `execHzn`. Intuitively, this will produce fewer constraints in the resulting MEMGRAPH. A kernel associated with a vertex cannot run until it has GPU RAM to write its output. If this GPU RAM is available very early in the simulation, then it gives the DISCO event processing loop more freedom to choose a vertex execution order that does not exactly match the simulated ordering, generating more opportunities to run available kernels while waiting for memory transfers.

The overall algorithm, BUILDMEMGRAPH, is given in Figure 10. Note that this variant of the algorithm assumes each tensor takes up exactly one slot in GPU RAM. In the "real life" case where tensors are variably-sized, the algorithm does not change appreciably—specifically, in the variably-sized case, freeing space for a tensor can evict a variable number of tensors to CPU RAM—but assuming uniformly-sized tensors simplifies the presentation.

```
simMalloc: Input: vertex v;                    simMallocForceReld: Input: vertex v;
Output: GPU memory slot for v                   Output: GPU memory slot for v
```

**find** open `slot` for $v$; **return** -1 if none     **remove** $v$ from `Evicted`
**return** `slot` if no previous occupant     `slot` $\leftarrow$ `simMalloc`$(v)$
$v' \leftarrow$ last owner of slot for $v$     **if** `slot` $\neq -1$ **then**
`Deps` $\leftarrow$ $\{v''$ s.t. edge $v' \rightarrow v'' \in$         **return** `slot`
TASKGRAPH$\}$     **end if**
**for** $v'' \in$ `Deps` **do**     **return** `simMallocOffld`$(v)$
    **add** edge $v'' \rightarrow v$ to MEMGRAPH
**end for**
**return** slot

`simMallocOffld`: **Input**: vertex $v$; **Output**: GPU memory slot for $v$

**find** GPU RAM `slot` for $v$ and determine victim (current occupant of `slot`) $v'$
**add** sequence $v' \rightarrow$ offload$_{v'} \rightarrow$ reload$_{v'}$ to MEMGRAPH
**add** edge offload$_{v'} \rightarrow v$ to MEMGRAPH
`Deps` $\leftarrow \{v''$ s.t. edge $v' \rightarrow v'' \in$ TASKGRAPH and $v''$ comes before $v$ in $V\}$
**for** $v'' \in$ `Deps` **do**
    **add** edge $v'' \rightarrow v$ to MEMGRAPH
**end for**
**rename** all instances of $v'$ in TASKGRAPH to reload$_{v'}$
**add** reload$_{v'}$ to `Evicted`
**return** slot

Figure 11: `simMalloc` variants used in MEMGRAPH construction.

At the highest level, the algorithm operates by first checking to see if it can allocate space for the vertex at the current allocation horizon, $v_{\texttt{allocHzn}}$. If it cannot, the algorithm makes sure there is space available for the output of the next vertex to be executed (the only way there is no space is if `allocHzn` = `execHzn` and the last allocation failed; this implies it is time to execute $v_{\texttt{execHzn}}$ and we just failed to allocate space for it). If there is space, the simulation "executes" $v_{\texttt{execHzn}}$.

There are four memory management subroutines used by the algorithm: three variants on `malloc` (`simMalloc`, `simMallocForceReld`, and `simMallocOffld`) and one variant on `free` called `simFree`. Like a traditional `malloc`, `simMalloc` finds an open slot for the allocation, but it also adds the memory dependencies to the MEMGRAPH necessary to ensure that the vertex $v$ that will occupy the slot will safely overwrite the previous occupant of the slot. `simMallocForceReld` is like `simMalloc`, but it is used in the case when a vertex must be reloaded because it is going to be used immediately, and hence the allocation cannot fail. `simMallocOffld` is a variant of `simMalloc` that cannot fail, as it finds a victim to offload to ensure the success of the allocation for vertex $v$, adding the `offload-reload` sequence to the MEMGRAPH. Crucially, it renames all instances of the victim $v'$ in the TASKGRAPH to refer to reload$_{v'}$. In this way, all "future" accesses to $v'$ will refer to its reloaded version. The routine also adds a memory dependency from the offload$_{v'}$ to $v$, as we cannot execute $v$ until offload$_{v'}$ has taken place, and freeing GPU RAM.

## 5 EXPERIMENTS

Our experiments evaluate the ability of DISCO to deal with Meta AI's LLaMA large language model (LLM) (Touvron et al., 2023), with severely constrained memory. We chose LLM training and inference as representative workloads. These tasks are computationally challenging and particularly difficult for modern ML systems given their large memory footprints. We assessed DISCO's performance on 7 billion and 65 billion parameter models.

Experiments were conducted on two machines: (i) an older $4 \times$ P100 GPU server (16 GB RAM each) and 22, 64GB DDR4 2666MHz CPU RAM modules, for a total of 1.3TB of RAM, and (ii) an Amazon Web Services `p4d.24xlarge` instance, equipped with eight A100 GPUs (40 GB RAM each) and 1.15TB of RAM. We were particularly interested in seeing the ability of DISCO to operate in a difficult environment with extremely limited GPU RAM, hence the P100 GPUs, with only 64 GB of GPU RAM total on the server. Key questions are: Can software help bridge the gap—particularly

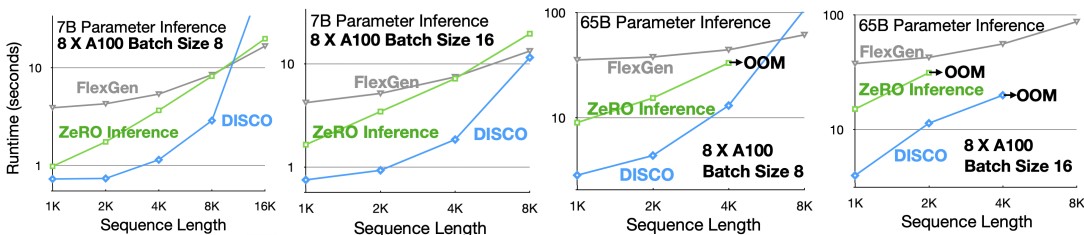

Figure 12: Time for LLaMA first token (prefill) inference, A100 server. "OOM" is out-of-memory.

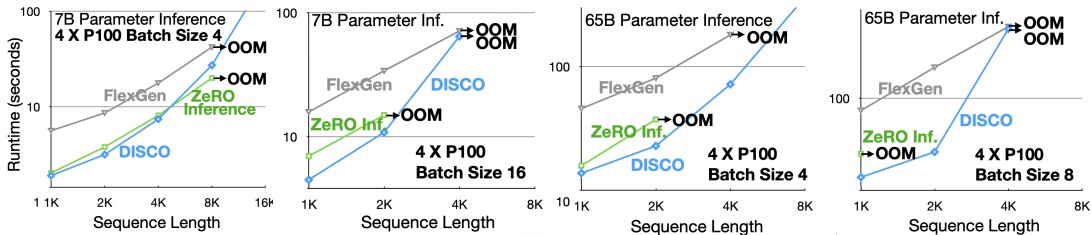

Figure 13: Time for LLaMA first token (prefill) inference, P100 server. "OOM" is out-of-memory.

the lack of GPU RAM—between older and newer hardware? Can DISCO facilitate model training and inference in a situation with limited RAM?

**(1) LLaMA first token inference.** Our experiments target "first token" inference (also known as "prefill"): How long does it take to produce the first output token, given an input prompt? We focus on prefill as it is exceedingly expensive in terms of the memory required, scaling quadratically with the size of the prompt. On both machines, we run DISCO, ZeRO Inference (Aminabadi et al., 2022) (using weight partitioning and model weight offload), and FlexGen (Sheng et al., 2023). For FlexGen, we use full CPU offload for activations. All testing is done using batched input, as batching is required for FlexGen and ZeRO (as DISCO simply runs a dataflow graph, it is agnostic to batching). For the smallest batch sizes considered, we test prefill input sequence lengths: 1K, 2K, 4K, 8K, and 16K tokens. For larger batches we use 1K, 2K, 4K and 8K. For DISCO, all model weights and computations were performed using 16-bit floating points, though FlexGen uses very low precision arithmetic to save RAM and speed compute. Results for the A100 GPU server are given in Figure 12. Results for the P100 GPU server are given in Figure 13.

One of the advantages of DISCO is that it executes arbitrary dataflow graphs in limited memory. Thus, as long as a computation is appropriately decomposed to run on multiple GPUs, DISCO can execute it. This means, for example, that DISCO does not need to perform inference over batches of input sequences and supports arbitrary combinations of model and data parallelism (unlike FlexGen and in ZeRO Inference). While batching tends to increase computational efficiency, the RAM used by a large batch means it is not possible to run inference over long sequences in limited memory (batching precludes that all 320GB of GPU RAM be dedicated to prefill for a single long sequence). To investigate the ability of DISCO to perform inference over a single long sequence, we test sequences sizes of up to 32K tokens, on both GPU servers and on both the 7B and 65B parameter models. Results are shown in Figure 15.

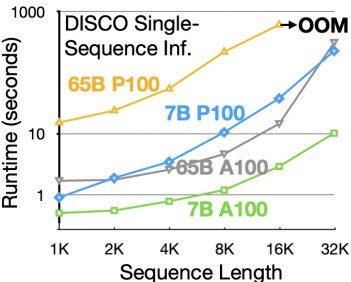

Figure 15: Single-sequence inference times.

**(2) LoRA training for LLaMA.** We also experiment with LoRA training (Hu et al., 2021). We use a LoRA rank of 16, and train LoRA adapters for the $K$, $V$, $Q$, and feedforward mapping matrices. Here we run DISCO and ZeRO Infinity (Rajbhandari et al., 2021); ZeRO is executed using all three "stages" (gradient partitioning, model weight partitioning, optimizer state partitioning) as well as CPU offload. Both DISCO and ZeRO use checkpointing during the forward pass to reduce the memory

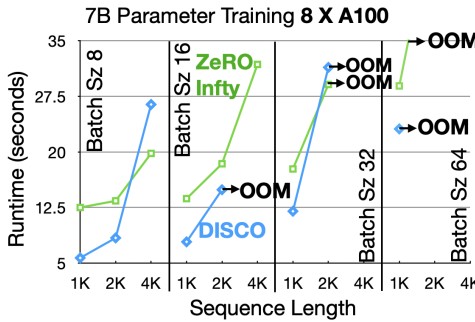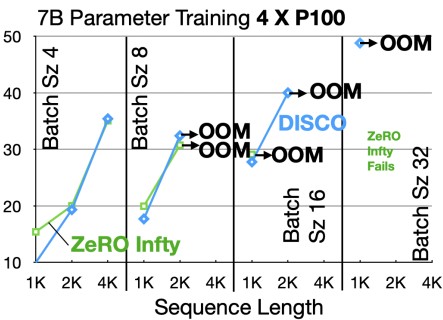

Figure 14: Comparing DISCO and ZeRO Infinity for LoRA training.

footprint. We measure the time it takes to run the forward and backward pass for one batch, with varying batch sizes and sequence lengths. All DISCO model weights are stored as single precision (32 bits). Results for training using both the P100 and the A100 server at 7B parameters are in Figure 14. Both systems had a difficult time training the 65B parameter model. DISCO was faster for the one case it was able to run (1K length sequence, batch size eight took 58.5 seconds using DISCO and 72.9 using ZeRO Infinity) but ZeRO Infinity was more robust to larger batch sizes, where DISCO failed.

**Discussion.** Throughout the first token inference experiments, DISCO typically performed the best in terms of latency, with ZeRO Inference generally much slower but outperforming FlexGen, except for larger sequence sizes. This would seem to validate the dataflow-based approach advocated for in the paper, at least if the goal is low latency.

To be fair, we note that FlexGen is designed for high throughput, as opposed to low latency, and FlexGen utilizes multiple GPUs only via pipelined parallelism. Note that FlexGen does not seem to get any slower when moving from batch size of eight to 16 on the A100 server. This suggests that filling the pipeline leads to substantial latency. Further, pipelined parallelism is more effective with more work in each pipeline stage, due to the high synchronization overhead and the need to try to overlap communication with computation, perhaps explaining FlexGen's better performance for larger sequences, which are more dense computationally.

ZeRO Inference takes a much different approach; it uses a highly synchronized form of model parallelism as it traverses the levels in a transformer, also carefully trying to overlap communication and computation, which may be more effective when there is more work at each level. DISCO, on the other hand, is radically different. It does not "understand" the levels in a transformer, does not need to synchronize processing of the various levels, and simply tries to asynchronously process kernels as fast as it can. If it is stuck waiting for communication, it seeks out other available tasks to execute.

For training, there were clear advantages of DISCO over ZeRO Infinity, especially for shorter sequence lengths. This was particularly true on the A100 server, where DISCO was often much faster. For batches of sequences of length 1K, DISCO often took less than 50% of the time to process each batch, compared to ZeRO (at a batch size of 8, the time to process 1K sequences was 5.6 seconds for DISCO and 12.5 seconds for ZeRO, for a batch size of 32 it was 12.1 seconds for DISCO and 17.8 seconds for ZeRO). The differences in performance were much less pronounced on the P100 server, though there DISCO was still faster. Finally, we note that both systems suffered significant out-of-memory errors during training. Interestingly, DISCO seemed to have more memory problems on the A100 server, whereas ZeRO Infinity had more problems with memory on the P100 server. We conjecture that some of that could be solved in DISCO with a better input dataflow graph, which cuts the input problem into smaller pieces.

## 6 CONCLUSIONS

In this paper, we introduced DISCO, a novel approach to CPU offload in ML workloads that leverages a work-conserving, dynamic scheduler to maximize resource utilization. By representing ML workloads as fine-grained dataflow graphs and asynchronously scheduling operations based on resource availability, DISCO effectively overlaps memory transfers and computation, mitigating the performance penalties associated with CPU offload and ensuring the resources are fully utilized.

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

## A  THE DISCO EXECUTION ENGINE

Once a MEMGRAPH has been produced, it is executed by the DISCO engine using a nondeterministic, event-based framework. As soon as a GPU is unused or a tensor is ready to be `offload`ed to RAM, the DISCO runtime can immediately assign any available work to the GPU or begin the transfer, without regard to the overall state of the computation. Also note that there are no calls to memory-management routines such as `cudaMalloc` or `cudaFree` during MEMGRAPH execution, as memory management is no longer dynamic. Tensor placement is pre-determined before execution, and if dependencies are respected, there can be no memory corruption due to race conditions.

Our execution engine consists of a central event loop that "launches" each vertex in the MEMGRAPH. A vertex can be launched when (1) all dependencies have been completed and (2) the required resources are obtained. When a vertex is launched, the corresponding operation is executed and then a provided callback is called to notify the event loop that the vertex has completed. In turn, the event loop frees up the obtained resources and keeps track of when vertices complete execution so that subsequent vertices can be launched. In practice, when launched, a vertex will execute one or more asynchronous CUDA operations on CUDA stream and will then call `cudaStreamAddCallback`. As such, every vertex requires as a resource a stream, where a single stream can only be used by a single launched vertex at a time. We use 5 streams per GPU. `offload`, `Reload` and inter-GPU communication vertices will call `cudaMemcpyAsync`. For CPU storage, we allocate a single, large contiguous block of memory with `cudaHostAlloc` with flags `cudaHostAllocPortable` and `cudaHostAllocWriteCombined`. When executing `Offload` vertices, we allocate into the CPU storage memory using our custom allocator; when executing `Reload` vertices, we free from our custom allocator. All compute vertices are executed using either cuTensor functions or hand-written CUDA kernels. An example where a hand-written CUDA kernel is beneficial is for executing portions of softmax so that less workspace memory and fewer vertices would be required. Two additional resources may be required for computing vertices: workspace memory as required for executing multiple cuTensor functions and locks around write-protected memory. As an example, we would execute a summation of $n$ tensors with $n$ calls to tensor increment sum-into kernels. However, the output memory would be protected by a resource so that only one sum-into can happen at a time. This implementation is designed to support non-determinism. We use CUDA version 11.8.0 and cuTensor version 2.0.1. All other code is C++.

## B  USE OF LARGE LANGUAGE MODELS

Large Language Models are used to polish the writing of this paper, as well as formatting algorithms and tables, but they do not contribute to a significant part of the paper. The intellectual content of this paper is the sole work of the authors.

