# OpenReview forum: "DISCO: Dynamic Scheduling for CPU Offload in ML Workloads"
_ICLR.cc/2026/Conference — Submitted to ICLR 2026_

### Official Review · Reviewer_bT7C · 2025-11-01

**Soundness:** 2
**Presentation:** 3
**Contribution:** 2
**Rating:** 4
**Confidence:** 3

**Summary:**

In this paper, DISCO targets memory-constrained ML workloads that rely on CPU offloading. It unifies computation and data movement through a fine-grained dataflow abstraction called TASKGRAPH, which models both computation and transfer dependencies. During compilation, DISCO constructs a MEMGRAPH that encodes data and memory dependencies and inserts offload or reload operations as needed. At runtime, a work-conserving dynamic scheduler executes ready tasks asynchronously once dependencies are satisfied and resources are available, covering GPU kernels as well as C2G and G2G transfers to maximize compute–communication overlap and prevent GPU idle time caused by batch-level synchronization. The system performs simulated execution to determine tensor memory placement in advance and eliminate overwrite conflicts, avoiding dynamic cudaMalloc and cudaFree during execution. Results show that DISCO achieves lower first-token latency than ZeRO-Inference and FlexGen in most settings, supports single-sequence contexts up to 32K tokens without batch parallelism, and outperforms ZeRO-Infinity in several training configurations, demonstrating strong usability and low-latency advantages under constrained GPU memory.

**Strengths:**

1. The fine-grained unified abstraction combined with a work-conserving scheduler significantly improves compute–communication overlap and resource utilization. Both theoretical examples and empirical results (e.g., first-token latency) show clear advantages over layer-wise batch-synchronized execution.

2. The compile-time construction of MEMGRAPH, which inserts memory dependencies and offload/reload operations, ensures correctness and parallelism. During execution, the system avoids dynamic GPU memory allocation, and its event-driven design with multiple CUDA streams minimizes scheduling and memory management jitter.

3. The system demonstrates strong practicality under extremely constrained GPU memory and diverse parallel paradigms. It supports single-sequence contexts of up to 32K tokens, remains insensitive to batch size, and outperforms ZeRO-Infinity in multiple LoRA training settings.

**Weaknesses:**

1. The evaluation primarily focuses on average latency and offline batch experiments, lacking a systematic analysis under realistic online workloads featuring bursty arrivals and mixed sequence lengths. In particular, there are no measurements of tail latency, continuous batching, stateful execution, or mechanisms such as rate limiting, prioritization, and preemption. Moreover, the study does not include end-to-end comparisons with recent mainstream inference stacks—such as vLLM and PagedAttention and FastGen under equivalent precision settings and operator stacks.

2. During compile-time MEMGRAPH construction, the paper illustrates the process mainly with equal-sized slot examples, without addressing how the system handles heterogeneous tensor size distributions, bandwidth and latency asymmetries, or how it mitigates victim selection imbalance, fragmentation, and reload-induced jitter under such conditions.

3. The paper also lacks systematic ablations that disable key components such as the work-conserving scheduler, fixed topological execution, or offload/reload insertion. As a result, it remains difficult to determine the primary sources of performance gains, whether the scheduler itself becomes a bottleneck at larger scales or higher concurrency, and where the system’s stable operating regime lies for larger batch sizes and model scales.

**Questions:**

1. How does the system preserve an acyclic dependency graph and safe‑overwrite invariants under variable-size tensors or multiple LoRA instances, while remaining compatible with continuous batching and paged KV caching mechanisms as in vLLM?

2. Please provide formal sufficient and necessary conditions for contention-free execution, along with upper bounds on compilation and allocation complexity. Clarify how victim selection mitigates jitter and thrashing under variable tensor sizes and heterogeneous interconnects, and specify the worst-case guarantees.

3. Under identical precision and operator/stack configurations, compare throughput and tail latency (e.g., P95/P99) against related work,  such as FastGen or others. Include ablation studies that disable work-conserving scheduling and offload/reload insertion.

---

> ### Author Response · Authors · 2025-11-26
>
> We thank the reviewer for the detailed and constructive assessment.
>
> ---
>
> ### **1. Online workloads, tail latency, and comparisons to vLLM / FastGen**
>
> DISCO is designed and evaluated as a **single-graph CPU-offload runtime**. The current prototype does **not** implement continuous batching, paged KV caching, dynamic arrivals, or tail-latency-oriented serving policies. Because these systems (vLLM, PagedAttention, FastGen) rely on specialized operator stacks, fused kernels, paging layouts, and batching logic that DISCO does not implement, a direct comparison under “identical stacks” is not feasible.
>
> ---
>
> ### **2. Variable-sized tensors and practical allocator behavior**
>
> The allocator in our implementation already supports **variable-sized tensors**. BUILDMEMGRAPH simply searches for a contiguous free range large enough for each tensor. Memory-dependency rules remain unchanged: a region is reused only after all prior reads complete, ensuring overwrite safety.
>
> Fragmentation is naturally reduced under tight memory because large allocations trigger evictions that consolidate free space. We experimented with compile-time defragmentation but observed little benefit. We will add a short clarification to the paper.
>
> ---
>
> ### **3. Ablations (scheduler, fixed ordering, offload/reload)**
>
> We agree that ablations help isolate the contributions of nondeterministic, work-conserving scheduling. In closely related prior work with the same runtime style, we implemented a partially deterministic variant (which prioritized contractions and large kernels) and observed the following average speedups for nondeterministic execution:
>
> - A100 inference: **4.02%**
> - P100 inference: **6.45%**
> - A100 training: **13.4%**
> - P100 training: **14.5%**
>
> These results show that allowing free ordering at runtime provides measurable benefit. A fully deterministic version is difficult to support in the current prototype due to how reductions are implemented.
>
> ---
>
> ### **4. Acyclicity, overwrite safety, and dynamic mechanisms (multi-LoRA, KV paging)**
>
> MEMGRAPH remains acyclic because all added memory dependencies respect the topological order of the TASKGRAPH. Overwrite safety is preserved by ensuring that a tensor is placed into a region only after all prior users of that region have completed.
>
> Multiple LoRA adapters simply introduce additional tensors and ops in the DAG; as long as the input TASKGRAPH is acyclic, MEMGRAPH remains acyclic.
>
> Dynamic mechanisms such as paged KV caching or continuous batching introduce **runtime graph changes**, which the current version of DISCO does not support. Extending DISCO to handle dynamic graph growth or multiple compiled segments is future work.
>
> ---
>
> ### **5. Conditions for correctness and compilation complexity**
>
> Correct execution requires two properties:
>
> 1. The TASKGRAPH encodes all true data dependencies and is acyclic.
> 2. MEMGRAPH adds memory dependencies so that each overwrite happens only after all prior reads finish.
>
> These conditions are sufficient to guarantee race-free execution under any valid runtime ordering.
>
> MEMGRAPH construction is a single linear pass over the topologically sorted TASKGRAPH plus a bounded search over active allocations. Compilation overhead is small in practice and amortized across many runs.
>
> ---
>
> ### **6. Contention-free execution and complexity**
>
> Our notion of **“contention-free’’** means: no valid execution can read from a GPU region after it has been overwritten. This is guaranteed under two conditions: (1) the TASKGRAPH is a finite DAG containing all true data dependencies, and (2) BuildMemgraph’s simulated allocator maintains exclusive GPU memory intervals and only reuses a region after inserting memory-dependency edges ensuring all consumers of the old contents execute before the reuse. Under these conditions, the MEMGRAPH remains acyclic, and any topological schedule is free of overwrite races.
>
> The compilation cost is one linear sweep over the TASKGRAPH (**O(n + m)**) plus allocator calls. With a simple ordered free-list or interval tree, each allocation/free/offload step costs **O(log n)**, giving a total construction cost of **O((n + m) + n log n)**. Each tensor can be evicted at most once between consecutive uses, so the number of offload/reload pairs is **O(m)**, and the MEMGRAPH size stays **O(n + m)**.
>
> Our victim-selection policy is correctness-oriented rather than performance-optimal. It guarantees bounded eviction frequency and race-freedom, but does not attempt to model heterogeneous PCIe/NVLink bandwidths.
>
> ---
>
> We hope that the clarifications provided above meaningfully address the concerns raised. If the responses resolve the uncertainties around the scope, design choices, and empirical behavior of DISCO, we kindly ask the reviewer to consider updating the score accordingly. We appreciate your thoughtful feedback and the opportunity to improve the paper.

---

### Official Review · Reviewer_gRdB · 2025-11-01

**Soundness:** 2
**Presentation:** 3
**Contribution:** 2
**Rating:** 2
**Confidence:** 5

**Summary:**

This paper introduces DISCO, a system that enables efficient overlap between communication and computation through asynchronous execution using a work-conserving dynamic scheduler. DISCO specifically targets CPU offloading scenarios where communication overhead is significant. The work-conserving dynamic scheduler is based on MEMGRAPH, which captures both data and memory dependences. MEMGRAPH is built upon a TASKGRAPH, which represents the dataflow of multi-GPU computation and can be generated by existing frameworks like FlexFlow or Alpa. By simulating the execution of the TASKGRAPH, MEMGRAPH is generated by adding offload/reload vertices when needed and inserting memory dependences to avoid race conditions in shared memory locations. Experiments on language models such as LLaMA-7B and LLaMA-65B, conducted on A100 and P100 GPU servers, show that DISCO outperforms FlexGen and ZeRO-Inference in first token inference, as well as ZeRO-Infinity in LoRA training.

**Strengths:**

The paper tries to address a significant challenge that arises when training models under limited memory conditions. Also, experiment results show that DISCO outperforms existing baseline systems in LLM inference and LoRA training.

**Weaknesses:**

The paper lacks a detailed explanation of how variable-sized tensors are handled. It merely states that the proposed algorithm, BUILDMEMGRAPH, does not change significantly for the “real-life” scenarios. However, without a detailed explanation, it remains unclear how the system addresses potential challenges such as fragmentation and simulation overhead that may arise in practical deployments.

 Also, the experiments are limited to a small set of workloads – only Llama is evaluated, the decode stage is not dealt with, and training is performed exclusively with LoRA. For the baseline systems, it would have been better to compare the proposed method with other general approaches mentioned in the Related Work section, such as pofo, AutoTM, and Checkmate. Finally, including an ablation study showing the time required to build MEMGRAPH through simulation would help demonstrate the practical usability of the system.

**Questions:**

-	The paper says that DISCO was proposed because it is difficult to engineer solutions for ML workloads that are not as simple. However, the evaluation uses transformer models, which generally exhibit relatively regular workloads. This raises a question of what the paper considers to be a “simple workload,” and further clarification on this criterion would be helpful.

-	DISCO’s runtime value for sequence length 16K is not seen clearly in Figures 12 and 13.

-	A more detailed analysis of the evaluation results would be desirable. For example, why was the performance difference between ZeRO Infinity and DISCO much less pronounced on the P100 server?

---

> ### Author Response · Authors · 2025-11-26
>
> We thank the reviewer for the constructive summary and for recognizing the significance of the problem and the strength of our results on both inference and training. We address the weaknesses and questions below.
>
> ---
>
> ### **1. Handling variable-sized tensors, fragmentation, and simulation overhead**
>
> Our simulated allocator supports variable-sized tensors. In practice, this modifies only the allocation step: instead of searching for a single slot, **BUILDMEMGRAPH** searches for a contiguous range of free space large enough for the tensor. Offload/reload vertices and memory-dependence insertion proceed exactly as in the fixed-size case. We will expand this description for clarity.
>
> Fragmentation is naturally mitigated under memory pressure: when a large tensor is allocated, it often triggers offloads of other tensors, which consolidates free memory.
>
> Simulation overhead is a **one-time offline cost**. A MEMGRAPH is reused for many iterations (training) or many requests (inference). The simulation is a single linear pass over the topologically sorted TaskGraph and is fast relative to total run time. For example, construction for 7B inference with batch size 8 on 4 GPUs takes **781.3 ms**.
>
> ---
>
> ### **2. Limited workloads and choice of baselines**
>
> We agree that general offloading approaches such as pofo, AutoTM, and Checkmate are relevant and part of the broader landscape. However, these methods are not designed or optimized for large-scale transformer inference or training. In contrast:
>
> - FlexGen and ZeRO-Inference are explicitly optimized for LLM inference under CPU offloading.
> - ZeRO-Infinity is specifically designed for training large transformer models when GPU memory is insufficient.
>
> Because our primary goal is to evaluate DISCO in the context of **LLM-scale CPU offloading**, we chose baselines that represent the strongest and most specialized systems in that setting. These LLM-specific approaches make much more aggressive use of architectural knowledge (e.g., KV-cache layout, layerwise structure, attention patterns), and thus provide a stricter and more meaningful comparison for demonstrating DISCO’s capabilities.
>
> LLaMA 7B and 65B already generate very large TaskGraphs (tens of thousands of vertices and dependencies), sufficiently stressing CPU offload engines. LoRA training was used because full-parameter training of 65B models is not feasible on our hardware for any method, baseline or otherwise.
>
> We agree that more models and additional baselines would further strengthen the paper, and we will incorporate these in future work.
>
> ---
>
> ### **3. Visibility of DISCO’s value at 16K sequence length (Figures 12 and 13)**
>
> We acknowledge the reviewer’s comment that DISCO’s advantage is harder to see at the far right of the plots. This is a visualization issue due to the scale of the plot. We will improve the figure formatting (e.g., zoomed-in insets or adjusted scaling) so the differences at 16K are easier to see. No data will be changed.
>
> ---
>
> ### **4. What is meant by a “simple workload”?**
>
> By “simple workloads,” we mean computations whose execution order and memory usage are predictable enough that static offloading plans work reliably—e.g., sequential pipelines or small models with minimal CPU–GPU transfer contention.
>
> Modern transformers, although regular in structure, produce **large, highly parallel TaskGraphs** with substantial, overlapping CPU–GPU transfers triggered by parameter and activation movement. These conditions make static offload plans brittle and motivate the dynamic, event-driven approach taken by DISCO. We will clarify this terminology in the revision.
>
> ---
>
> ### **5. Why DISCO vs. ZeRO-Infinity differences are smaller on P100**
>
> On P100 servers, GPU compute is slow and the compute-to-transfer ratio is lower. In this regime, long-running kernels provide abundant time for transfers to complete, so all systems benefit from natural overlap. As a result, DISCO’s fine-grained scheduling provides a smaller relative improvement.
>
> On A100, compute is much faster, transfers represent a larger fraction of wall time, and DISCO’s aggressive overlap yields a more pronounced advantage. We will add this explanation to the evaluation section.
>
> ---
>
> We sincerely thank the reviewer for the time and care invested in evaluating our work. We hope that the clarifications provided above meaningfully address the concerns raised. If the responses resolve the uncertainties around the scope, design choices, and empirical behavior of DISCO, we kindly ask the reviewer to consider updating the score accordingly. We appreciate your thoughtful feedback and the opportunity to improve the paper.

---

### Official Review · Reviewer_j77t · 2025-11-01

**Soundness:** 3
**Presentation:** 3
**Contribution:** 4
**Rating:** 8
**Confidence:** 4

**Summary:**

This paper introduces DISCO, a dynamic schedulng tool for CPU offloading of ML algorithms. DISCO helps address the issue of OOM errors, by better utilizing and pipelining information across the CPU and GPU, by constructing a memgraph and using for scheduling. The results are tested on both modern (A100) and older (P100) GPUs, which showcase its generality and how it can even resurrect older GPUs with limited memory to be practical for running LLMs.

**Strengths:**

+ Design of the Memgraph and maintaining consistency is a challenge, and the authors did a good job to simplifiy and maintain dependencies while addressing the core memory challenge.
+ I really liked the experiment of running a 7B param model on P100. It actually helps highlight the benefits of DISCO.

**Weaknesses:**

- The technique isn't novel per se. But that is less important given the benefits provided and the memory wall we are hitting with AI these days.

**Questions:**

- Will this be open source?
- How is this related to CPU pipelining? Could a scoreboard-like technique be used for DISCO's scheduling management?

---

> ### Author Response · Authors · 2025-11-26
>
> We thank the reviewer for the positive and thoughtful assessment. We are glad that the MemGraph design, consistency guarantees, and the P100 experiments were found valuable. We address the reviewer’s questions below.
>
> ---
>
> ### **Q1. Will DISCO be open source?**
>
> Yes. We intend to open-source DISCO after the review process concludes. The implementation currently contains internal tooling that must be cleaned up, but our plan is to release the system publicly.
>
> ---
>
> ### **Q2. Relation to CPU pipelining / scoreboarding**
>
> Scoreboarding is a hardware mechanism used in classic CPUs to track hazards for a single linear instruction stream. A scoreboard monitors a small number of functional units and issues an instruction when it is safe to do so, based on read/write hazards and functional-unit availability.
>
> DISCO’s runtime is different in several important ways:
>
> - DISCO operates over **thousands of fine-grained operations** in an explicit dependency DAG, not a linear instruction stream.
> - Scheduling legality depends not only on data dependencies but also on **GPU-memory residency**, **offload/reload dependencies**, and **resource constraints** (streams, locks, workspace).
> - Transfers and kernels execute **asynchronously** and may complete in a nondeterministic order.
> - The scheduler is fully **work-conserving**: any dependency-ready operation may execute immediately.
>
> A crucial distinction is that the **TaskGraph is constructed such that any topologically valid execution order produces the same correct output**. All true data dependencies are explicitly encoded in the graph. Therefore, DISCO’s scheduler can freely pick among any ready vertices without risking correctness. Classical scoreboarding does not guarantee correctness under arbitrary reordering; it only prevents hazards within a fixed instruction stream.
>
> Thus, the connection to scoreboarding is purely conceptual (executing an operation when it becomes “ready”), while DISCO generalizes this idea to a heterogeneous, asynchronous, large-scale DAG with explicit memory-correctness constraints.
>
> ---
>
> We thank the reviewer again for their constructive comments and are pleased that they found the contribution significant.

---

### Official Review · Reviewer_KCDr · 2025-11-01

**Soundness:** 2
**Presentation:** 2
**Contribution:** 2
**Rating:** 4
**Confidence:** 3

**Summary:**

This paper proposes DISCO, a system to address performance bottlenecks from CPU offloading in ML workloads. The core contribution is a "work-conserving" dynamic scheduler that operates on a pre-compiled, fine-grained dependency graph MEMGRAPH. By scheduling tasks asynchronously whenever resources are free, DISCO aims to maximize the overlap between computation and I/O, thus improving overall resource utilization.

**Strengths:**

- The fine-grained, dynamic, "work-conserving" scheduler is an advancement over the static, coarse-grained pipelines used by current SOTA systems.

- The MEMGRAPH-based approach is model-agnostic. It can handle any workload that can be expressed as a dataflow graph, making it general.

- The paper provides empirical evidence that DISCO achieves significantly lower latency than SOTA systems in key tasks like LoRA training and first-token inference.

**Weaknesses:**

- DISCO "failed" during the 65B model training, whereas the baseline (ZeRO Infinity) was "more robust". This raises significant questions about the practical feasibility and stability of this complex scheduling approach at scale.

- The MEMGRAPH is generated statically before execution. This design seems ill-suited for dynamic workloads, particularly inference, where runtime bottlenecks like the dynamically growing KV Cache(as identified by work like TightLLM) change with every iteration. The paper does not adequately address how this static plan would adapt.

**Questions:**

- Can you elaborate on the root cause of the 65B model "failure"? Does it stem from an inherent scalability bottleneck such as memory management in the dynamic scheduler itself?

- How does your static MEMGRAPH design handle runtime-dynamic bottlenecks, such as the growing KV Cache in inference, where the I/O load changes with every iteration?

- The paper's optimizations are focused entirely on the GPU-CPU RAM (Tier-2) bottleneck. However, for extreme-scale models, the true bottleneck may lie at Tier-3 (CPU-SSD). It would be interesting to know if DISCO's PCIe scheduling optimizations would provide any meaningful benefit when the entire system is bottlenecked by much slower storage I/O. What benefit would you expect your system to provide in a Tier-3 offloading scenario, where the primary bottleneck is orders of magnitude slower than the one you are optimizing?

---

> ### Author Response · Authors · 2025-11-26
>
> We thank the reviewer for the constructive assessment and for highlighting the generality of the MEMGRAPH approach and the empirical gains on both inference and training. We address the weaknesses and questions below.
>
> ---
>
> ### **1. Root cause of the 65B training “failure”**
>
> The 65B training failure does *not* stem from DISCO’s scheduler, memory-dependency logic, or any instability in the dynamic execution. The underlying issue is the **coarseness of the TaskGraph decomposition**, not the scheduling approach itself.
>
> In our current decomposition, several kernels—most notably attention—have working sets that already nearly saturate GPU memory. When a single vertex’s working set exceeds available GPU RAM, no offload strategy can succeed unless the computation is decomposed further.
>
> DISCO can always execute a more fine-grained TaskGraph, but such decomposition increases CPU–GPU traffic and hurts performance. In this work, we chose a practical decomposition that reflects common practice rather than one that shards operations solely to avoid OOM.
>
> Importantly, the TaskGraph is constructed so that **any topologically valid execution order produces the same correct output**. All true dependencies are explicit, so the work-conserving scheduler is free to launch any ready vertex without affecting correctness. Thus, nondeterministic scheduling is safe by construction and is not related to the 65B failure.
>
> ---
>
> ### **2. Static MEMGRAPH vs. dynamic workloads (e.g., growing KV cache)**
>
> This is a valid concern. The current version of DISCO targets workloads whose computational structure and memory footprint per iteration are known ahead of time (e.g., first-token inference, forward/backward passes).
>
> Dynamic structures such as a growing KV cache would change the dataflow graph over time. The current system does not yet handle this natively.
>
> Two practical approaches can be supported:
>
> 1. **Multiple precompiled graphs**
>    Partition inference into stable regions (prefill, early decode, late decode). Each region uses a separate MEMGRAPH, and a lightweight controller switches between them.
>
> 2. **External control flow around static MEMGRAPH segments**
>    The core per-step operations (matmuls, attention mechanisms) are static, while only the KV-cache access pattern grows. A small controller can handle KV-cache management and invoke the appropriate static MEMGRAPH segments.
>
> Supporting full incremental graph growth is future work; we will clarify this scope.
>
> ---
>
> ### **3. Tier-3 (SSD) offloading and expected benefit of DISCO**
>
> SSD-based offloading introduces latencies and bandwidth limits that are orders of magnitude worse than Tier-2 (CPU RAM). DISCO’s PCIe/NVLink scheduling cannot eliminate these fundamental device limits.
>
> However, DISCO would still be beneficial in a Tier-3 setting:
>
> - maximizing overlap between SSD→CPU, CPU→GPU, and GPU compute,
> - avoiding avoidable GPU idle periods,
> - handling transfer-time variability via event-driven scheduling,
> - and exploiting fine-grained memory dependency information to plan offload order.
>
> That said, when SSD is the dominant bottleneck, the expected improvement is **reduced idle time**, not elimination of the bottleneck. We agree this scenario is important and will note it more clearly.
>
> ---
>
> We appreciate the reviewer’s constructive feedback and believe these clarifications strengthen the framing of the system’s scope and limitations.

---

### Official Review · Reviewer_TxhG · 2025-11-02

**Soundness:** 1
**Presentation:** 1
**Contribution:** 1
**Rating:** 2
**Confidence:** 4

**Summary:**

DISCO introduces a runtime and scheduling approach for memory-constrained ML workloads that rely on CPU offloading. Instead of processing models layer-by-layer, DISCO represents computations as a fine-grained dataflow graph and uses a work-conserving dynamic scheduler to overlap CPU–GPU transfers and GPU compute, to ensure that resources are not idle.

**Strengths:**

Efficient CPU offloading is an important challenge

**Weaknesses:**

* Difference between standard double buffering unclear. The need for overlapping between compute and communication is a very well-known systems optimization. Figure 3 is misleading because it is a highly unoptimized version of how CPU offloading should work. It is unclear what the novelty of the proposed system is from the paper.
* The Related work simply lists a set of prior works but does not clearly articulate what shortcomings these works all have that the proposed paper addresses.
* There is no motivation section that demonstrates in state-of-art ML frameworks that this underutilization of resources actually occurs.
* There is only one model evaluated (LLAMA). This is insufficient for a work that aims to address a system-level bottleneck.
* It is unclear what the baseline serving framework is and the need for CPU-offloading is also not clear in the baseline system. Does it have insufficient memory resources for inference and fine-tuning?
* It is not clear what a "level" of a neural network is. Does this mean a layer?

**Questions:**

Please see under weaknesses.

---

> ### Author Response · Authors · 2025-11-26
>
> We thank the reviewer for their feedback and address the concerns below.
>
> ---
>
> ### **1. “Difference between standard double buffering unclear; Figure 3 misleading; novelty unclear.”**
>
> Figure 3 is only an illustration to show the benefit of overlapping communication and computation. It is not intended to represent an optimized or complete implementation. The contribution of DISCO is not double buffering; the novelty is the combination of:
>
> - treating the entire computation as a **fine-grained TaskGraph**,
> - compiling it into a **MemGraph** with explicit data and memory dependencies and no race conditions, and
> - executing it using a **work-conserving, asynchronous, non-layer-synchronous scheduler**.
>
> This allows DISCO to schedule any ready operation as soon as its dependencies and resources permit, without synchronization barriers. Existing CPU-offload systems either (1) require architectural assumptions (e.g., LLM layer structure, KV-cache layout), or (2) rely on bulk-synchronous execution. DISCO demonstrates that a generic, dependency-driven, work-conserving approach can improve utilization without model-specific scheduling rules.
>
> ---
>
> ### **2. “Related work does not articulate shortcomings of prior work.”**
>
> The key shortcomings addressed by DISCO are:
>
> - **General offload systems** (e.g., SwapAdvisor, AutoTM, pofo) typically operate on static graphs and do not use a work-conserving runtime; they do not exploit asynchronous overlap when CPU transfers behave unpredictably.
> - **LLM-specific systems** (e.g., FlexGen, ZeRO) rely on careful model-specific analyses (layer structure, KV-cache behavior, prefetch rules) and introduce synchronous execution stages. DISCO avoids architectural assumptions and works directly from a generic dataflow graph.
>
> ---
>
> ### **3. “No motivation section showing underutilization occurs in real frameworks.”**
>
> Synchronous execution stages naturally lead to underutilization: in bulk-synchronous frameworks, all kernels for a layer must wait until all required weights arrive, and transfers cannot overlap with compute across layer boundaries. DISCO removes these synchronization constraints by allowing any dependency-ready vertex to execute immediately.
>
> ---
>
> ### **4. “Only one model evaluated (LLaMA). Insufficient.”**
>
> LLaMA 7B and 65B were chosen because they generate very large TaskGraphs when decomposed across multiple GPUs and are representative of modern transformer workloads. These models already produce thousands of kernels and tens of thousands of dependencies per inference, which stress CPU-offload systems significantly. We agree that additional models would further strengthen the paper.
>
> ---
>
> ### **5. “Baseline serving framework unclear; need for CPU-offloading unclear.”**
>
> Both hardware platforms used (P100 and A100) encounter memory limitations for LLaMA with long sequences or training workloads. The baselines (FlexGen and ZeRO) also rely on CPU offload under these conditions.
>
> ---
>
> ### **6. “Unclear what a ‘level’ of a neural network is.”**
>
> We will replace “level” with “layer” for clarity. The intention was simply to refer to successive layers of the transformer block stack.
>
> ---
>
> We sincerely thank the reviewer for the time and care invested in evaluating our work. We hope that the clarifications provided above meaningfully address the concerns raised. If the responses resolve the uncertainties around the scope, design choices, and empirical behavior of DISCO, we kindly ask the reviewer to consider updating the score accordingly. We appreciate your thoughtful feedback and the opportunity to improve the paper.

---

### Official Review · Reviewer_Hcji · 2025-11-03

**Soundness:** 2
**Presentation:** 2
**Contribution:** 2
**Rating:** 2
**Confidence:** 4

**Summary:**

The paper introduces DISCO, a novel system for dynamic scheduling of CPU offload in machine learning workloads. Unlike traditional bulk-synchronous approaches, DISCO models ML workloads as fine-grained dataflow graphs and uses a work-conserving dynamic scheduler to overlap memory transfers and computation. The key contribution of the paper is introducing MEMGRAPH and the corresponding algorithm for constructing MEMGRAPH, a DAG that includes data dependency and memory dependency to allow dynamic scheduling along with correctness. The proposed approach demonstrates a significant speedup over existing approaches for inference in memory-constrained systems.

**Strengths:**

1. The paper focuses on an important problem: improving resource utilization for machine learning workflows by improving the overlap of CPU-GPU communication and GPU operations.
2. The proposed approach of using a DAG with all dependencies captured is a generalized approach and can be used for different types of models.
3. The paper is well written, and the explanation of MEMGRAPH construction is easy to follow
4. The algorithm proposed for MEMGRAPH construction is lightweight.

**Weaknesses:**

1. The interaction of proposed dynamic scheduling with different GPU parallelism strategies (e.g., model parallelism), which might require synchronization after each block, is not clear.
2. The paper proposes to use a dynamic work-conserving scheduler, but does not provide much detail on it. Overheads of graph construction and scheduling are unclear as well
3. DISCO faces more OOM compared to existing work, hence its usefulness for training is unclear

**Questions:**

I have the following questions for the authors:
1. Impact of model parallelism: Several model parallelism strategies require GPU synchronization at the end of the layer, which might limit the scope and impact of the proposed asynchronous execution. Furthermore, I could not find details on what parallelism strategy is used for the evaluation section, which makes it hard to evaluate the interaction with parallelism strategies.
2. Example given in Figure 3 does not seem to account for the required synchronization before executing layer 2 for the tensor parallelism strategy.
3. Can you provide more details on the dynamic work-conserving scheduler? What is the overhead of this dynamic scheduler? Does it make scheduling decisions at runtime (*dynamic* work might be confusing here) or at compile time?
4. How does the size of this MEMGRAPH and the overhead of constructing/scheduling increase with the model size and the cluster size (e.g., training with 1024 GPUs)?
5.  Can you provide more details on the training results? The paper mentions that ZeRO Infinity was more robust to larger batch sizes and DISCO fails. It is unclear why DISCO results in OOM more often than the prior work, and how DISCO can be improved for training.

---

> ### Author Response · Authors · 2025-11-26
>
> We thank the reviewer for the positive feedback on the significance of the problem, the generality of the approach, and the clarity of the MEMGRAPH construction. We address all weaknesses and questions concisely below.
>
> ---
>
> ### **W1 / Q1–Q2. Interaction with model parallelism and synchronization**
>
> DISCO does not impose any parallelism strategy. It executes the computation exactly as expressed in the **TaskGraph**. Any required synchronization in a chosen parallelism scheme (e.g., end-of-layer collectives) appears as explicit dependencies in the TaskGraph and is therefore automatically enforced in the MemGraph. DISCO never violates such dependencies.
>
> The TaskGraphs in our experiments come from an external decomposition algorithm we developed. Importantly, this decomposition **does not introduce artificial synchronization stages** — only true data dependencies are encoded. This is beneficial for DISCO: unnecessary synchronization would reduce the scheduler’s ability to overlap CPU–GPU transfers with compute.
>
> Figure 3 is meant as an illustration of overlap. In full models, any required synchronization simply appears as dependencies that DISCO respects.
>
> ---
>
> ### **W2 / Q3. Details and overhead of the dynamic work-conserving scheduler**
>
> All memory-placement choices — allocation, offload/reload insertion, and memory-dependency edges — are decided **offline** during MemGraph construction. During execution, there are **no dynamic GPU allocations**, and the runtime overhead (dependency counters + event callbacks) is negligible relative to kernel execution and transfers.
>
> DISCO’s runtime scheduler is fully **dynamic** and **event-driven**:
>
> - A vertex launches as soon as its dependencies are satisfied and its required resources (GPU stream, locks, workspace) are available.
> - Operations run asynchronously and signal completion via callbacks.
> - The scheduler is **work-conserving**: if a legal operation exists for an idle resource, it is executed immediately.
>
> ---
>
> ### **W2 / Q4. MemGraph size and compilation overhead**
>
> MemGraph size equals the TaskGraph plus additional offload/reload vertices and memory-dependency edges when memory is constrained, which grows linearly. The construction algorithm performs a single pass over the topologically sorted TaskGraph with a simulated allocator. This is a one-time offline cost for a given model and hardware configuration, amortized over many inference or training runs.
>
> The simulation is a single linear pass over the topologically sorted TaskGraph and is relatively fast compared to the total runtime. For example, MemGraph construction for 7B inference with batch size 8 on 4 GPUs takes **781.3 ms**.
>
> Extending the approach to multi-node clusters would require a cross-machine-aware decomposition algorithm and can be done fairly easily.
>
> ---
>
> ### **W3 / Q5. OOM behavior in training**
>
> In LoRA training, DISCO is often substantially faster than ZeRO-Infinity when both systems run successfully. Both systems encounter OOMs. DISCO’s failures arise from the **coarseness of the TaskGraph decomposition**, not from limitations in the scheduler or MemGraph.
>
> Our decomposition may produce large kernels whose working sets nearly saturate GPU memory. A finer-grained decomposition would eliminate more OOMs — DISCO can always execute more granular graphs — but this would substantially increase CPU–GPU traffic and harm runtime performance. We therefore use a decomposition that reflects realistic practice rather than an artificially fine-grained one chosen solely to avoid OOM.
>
> DISCO could utilize larger kernels and improve Model Flops Utilization (MFU) during training. We note our training speed of the 7B model with A100:
>
> ---
>
> ## Batch Size 8
>
> | Model | 1K  | 2K  | 4K  |
> |-------|-----|-----|-----|
> | DISCO | 5.7 | 8.4 | 26.4 |
> | ZeRO  | 12.5 | 13.4 | 19.8 |
>
> ---
>
> ## Batch Size 16
>
> | Model | 1K   | 2K   | 4K  |
> |-------|------|------|------|
> | DISCO | 7.87 | 14.9 | OOM  |
> | ZeRO  | 13.7 | 18.4 | 31.8 |
>
> ---
>
> ## Batch Size 32
>
> | Model | 1K   | 2K   | 4K  |
> |-------|------|------|------|
> | DISCO | 12.0 | 31.4 | OOM  |
> | ZeRO  | 17.7 | 29.1 | OOM  |
>
> ---
>
> ## Batch Size 64
>
> | Model | 1K   | 2K    | 4K  |
> |-------|------|-------|------|
> | DISCO | 23.1 | OOM   | OOM  |
> | ZeRO  | 28.9 | 52.62 | OOM  |
>
> ---
>
> We sincerely thank the reviewer for the time and care invested in evaluating our work. We hope that the clarifications provided above meaningfully address the concerns raised. If the responses resolve the uncertainties around the scope, design choices, and empirical behavior of DISCO, we kindly ask the reviewer to consider updating the score accordingly. We appreciate your thoughtful feedback and the opportunity to improve the paper.

---

### Author Response · Authors · 2025-12-04
**Summary of Author Response for AC Consideration**

We thank the reviewers and AC for their time. Below we briefly synthesize the key parts of the reviews and our rebuttal.

**Technical contribution and novelty.**
The contribution is *not* simply overlapping compute and communication. Instead, DISCO provides:

1. A **generic TASKGRAPH → MEMGRAPH compilation** that encodes both data and memory dependencies and guarantees that *any* topological schedule is race-free and yields the same outputs.

2. A **work-conserving, event-driven runtime** that executes arbitrary ready vertices (kernels and transfers) without layer- or stage-synchronous barriers.

Existing CPU-offload systems either (a) are general but static and non–work-conserving, or (b) are **LLM-specific**, with hand-engineered schedules, paging layouts, and synchronization stages (e.g., ZeRO-{Inference,Infinity}, FlexGen). DISCO is, to our knowledge, the first system to show that a **generic, dependency-driven, work-conserving** runtime can match and often beat these specialized LLM-specific systems in the LLM regime.

**Soundness and overhead.**
All memory placement and offload/reload decisions are made **offline** during MEMGRAPH construction via a single pass over a topologically sorted TASKGRAPH with a simulated allocator. For example, MEMGRAPH construction for LLaMA-7B inference with batch size 8 on 4 GPUs takes **781.3 ms**, and this cost is amortized over many inference requests or training iterations.

The runtime scheduler itself is simple (dependency counters + callbacks) and introduces negligible overhead relative to kernel execution and transfers. We also stated formal conditions for “contention-free” execution: a finite acyclic TASKGRAPH that contains all true data dependencies, plus a MEMGRAPH that only reuses a GPU region after inserting memory-dependency edges that ensure all prior reads of that region complete. Under these conditions, any topological schedule is overwrite-race-free.

**OOM behavior and 65B training.**
We clarified that the 65B training failures stem from the **coarseness of the TaskGraph decomposition** (single kernels whose working sets exceed GPU RAM), not instability or incorrectness of DISCO’s scheduler or memory logic. DISCO can always execute a more fine-grained TaskGraph; in this work we intentionally use a realistic “large-kernel” decomposition rather than artificially sharding operators solely to avoid OOM. On configurations where both systems run, DISCO is often substantially faster than ZeRO-Infinity (e.g., up to ~2× speedup on 7B LoRA training).

We hope these clarifications help distinguish the core technical contribution and address the main soundness and robustness concerns raised in the reviews.

---

### Meta-Review · Area_Chair_Dme8 · 2026-01-11

**Summary:**

There are 6 reviewers provided a wide range of comments about this submission, and they raised a set of consistent concerns about the novelty, limitation, and experimentation of this work. The authors did provide some answers to the comments, but it didn’t seem to have addressed the reviewers concerns with convincing evidence.

Given the low ratings from most reviewers, I suggest a reject for this submission.

**Reviewer Concerns:**

The 6 reviewers have provided a wide range of comments about this submission, such as:
•	Concerns about the novelty of the work
•	The overhead not clear, especially with synchronization issues
•	Open-source or not
•	Concerns about the used metrics (avg latency) and workloads (offline vs realistic online workloads), and no end-to-end comparison
•	The workload graph may be too simplistic (homogenous vs heterogeneous)
•	Lacks systematic ablation with insufficient exp comparisons
•	Lacking some technical details (such as how to preserve an acyclic dependency)
•	Lack of theoretic analysis (sufficient and necessary conditions for contention-free execution)

The authors have also provided their answers to those comments. But their answers didn’t seem to be sufficient to generate enough enthusiasm from reviewers. The main concerns are about the novelty of the work, the limited applicability of the work, and not enough comparison to state-of-the-art with more careful/convincing experimentation.

**Reviewer Scores:**

None of the reviewers would change their rating based on the answers the authors have provided.

---

### Decision · Program_Chairs · 2026-01-26

Reject